# Sporotrichosis Outbreak Due to *Sporothrix brasiliensis* in Domestic Cats in Magallanes, Chile: A One-Health-Approach Study

**DOI:** 10.3390/jof9020226

**Published:** 2023-02-09

**Authors:** Pamela Thomson, Carlos González, Olivia Blank, Valentina Ramírez, Camila del Río, Sebastián Santibáñez, Pamela Pena

**Affiliations:** 1Laboratorio de Microbiología Clínica y Microbioma, Escuela de Medicina Veterinaria, Facultad de Ciencias de la Vida, Universidad Andrés Bello, Santiago 8370134, Chile; 2Laboratorio de Anatomía e Histopatología, Escuela de Medicina Veterinaria, Facultad de Ciencias de la Vida, Universidad Andrés Bello, Santiago 8370134, Chile; 3Laboratorio de Histopatología, CITOVET, Ñuñoa, Santiago 7750538, Chile; 4Clínica Veterinaria Timaukel, Punta Arenas 6210648, Chile

**Keywords:** *Sporothrix brasiliensis*, cats, outbreak, zoonotic disease

## Abstract

Sporotrichosis is an implantation mycosis with subcutaneo-lymphatic or, more rarely, a viscerally disseminated affection; it can be acquired through traumatic percutaneous inoculation of the fungus present in soil or plant matter, or by feline scratching. Among the causative agents, *Sporothrix brasiliensis* is considered the most virulent species with a high prevalence in Brazil and recently in Argentina. Objective: To describe a *S. brasiliensis* outbreak in domestic and feral cats detected in the Magallanes region of southern Chile. Materials and Methods: Between the months of July and September 2022, three cats presented with suppurative subcutaneous lesions located mainly on the head and thoracic limbs. The cytology revealed the presence of yeasts with morphological characteristics suggestive of *Sporothrix* spp. The histopathology confirmed pyogranulomatous subcutaneous lesions associated with the presence of the same yeasts. The fungal culture followed by the partial gene sequence and analysis of the ITS region confirmed the diagnosis of the *S. brasiliensis* as the causative agent. The cats were treated with itraconazole associated in one case with potassium iodide. The evolution of the patients was favorable in all cases. Conclusions: An outbreak caused by *S. brasiliensis* was detected in domestic and feral cats in austral Chile. The correct identification of this fungus and antifungigram is essential for treatment decisions and for designing dissemination control and prevention programs under a one health approach that consider the health of people, animals, and the environment.

## 1. Introduction

Sporotrichosis is frequently a cutaneous or cutaneous-lymphatic mycosis of a zoonotic nature and of worldwide distribution [1,2,3]. Most cases result from the traumatic inoculation of propagules of *Sporothrix* spp. in the host cutaneous tissue. This is considered an implantation mycosis that can occur through an animal or environmental transmission route [4,5,6,7]. *Sporothrix* spp. is a dimorphic fungus, which, in its saprophytic phase, develops a filamentous form, mainly in soils rich in organic matter, at 25 °C; while at 37 °C, in the host tissues, it remains as yeast, generating infection; both phases of the fungus are culturable in the laboratory [8,9,10,11,12].

Currently, 53 *Sporothrix* species have been described [13], which are divided into two clades, one is “clinical or pathogenic”, isolated from human or animal cases; composed of *S. schenckii* of worldwide distribution, *S. globosa* is present in Argentina, Venezuela, United Kingdom, Spain, Italy, China, Japan, USA, and India [14,15,16,17,18,19,20,21,22,23,24]. *S. luriei* is present in South Africa and *S. brasiliensis* is restricted to South America [14,15,16,17,18,19,20,21,22,23,24]. The remaining species are nested in an “environmental clade”, often associated with organic substrates such as soils with decaying plant remains, wood, or insects [25,26,27,28].

Advances in research on the clinical/pathogenic species have allowed significant information to be generated in terms of morphology [29], physiology [30], genetics [31], epidemiology [32], and virulence [22,33,34,35,36], among other aspects, which has facilitated their identification.

Within the pathogenic species, *S. brasiliensis* is categorized as the most virulent, becoming the main etiological agent of zoonotic transmission between cats and humans [37,38]. Transmission can occur through scratches, bites, or contact with exudates from skin lesions of infected cats [39,40,41,42,43]. In addition, cats can carry the fungus in their claws, which may have an important role in the epidemiology of the disease [42,44,45].

A recently published study by Morgado et al. [46] mentions that South America is the continent with the highest animal sporotrichosis prevalence (81%), followed by Asia and Europe [47], while North America and Africa report the same proportion. In South America, most of the isolates are concentrated in Brazil [48] and then in Argentina [38] and occur mainly in cats, followed by dogs [49,50,51]. The most prevalent species on the continent is *S. brasiliensis,* and *S. schenckii* appears less frequently [46,52].

The development of the disease is related to predisposing factors such as exposure to the infectious agent, the depth of the wound, and the immune response the host [9,53,54]. The incubation period varies from 13 days to 3 months, mainly affecting the skin and lymph nodes adjacent to the lesion. In cats, the most frequent clinical signs are associated with the skin presentation, characterized by erosions, ulcers, and fistulas accompanied by serous-bloody or purulent exudate, mainly located on the head, face, neck, and in the distal part of the extremities [37,50,55,56,57,58]. The extra-cutaneous form of pulmonary, ocular, or neurological localization is seen less frequently [37,53,59,60,61,62]. The highest incidence of the disease occurs in out door non-castrated male cats in constant contact with the outside world; its habit of scratching the vegetation facilitates the spread of the fungus in the environment [6,63]. In addition, its territorial and fighting behavior with other cats allows rapid transmission between individuals of this species [57,64]. As a treatment, itraconazole alone or associated with potassium iodide continues to be the first option against this type of infection [65,66,67,68,69].

We describe an outbreak by *S. brasiliensis* in cats from Patagonia in the southern part of Chile, which is the first isolation report of this agent in cats in our country.

## 2. Materials and Methods

### 2.1. Geographic Location

The outbreak that we describe below occurred between the months of July and September 2022 in the Magallanes Region, in southern Chile. Cases 1 and 2 come from the Río Verde County (53°09′46″ S 70°54′29″ W), a small rural village with approximately 250 people. Case 3 comes from the city of Punta Arenas (52°36′28″ S 71°30′28″ W) with 143,000 inhabitants (www.bcn.cl). The distance between both points is 98 km.

### 2.2. Case Description

Case 1: Feline, female, domestic long hair (DLH), 1,2 years, 1.5 kg, wild behavior, lived outdoors with a group of cats in the Río Verde location and had no owner. Due to the skin lesions she had and the presumptive diagnosis of a fungal non-identified disease, the local veterinarian prescribed itraconazole (Itraskin, Drag Pharma, Santiago, Chile). After observing that her multiple wounds did not heal, she was taken to a veterinary center located in the city of Punta Arenas (Clinica Veterinaria Timaukel).

On examination, she had sero-bloody subcutaneous wounds located mainly on the face and forelimbs, with laceration of the left ear; she was also very thin and weak. A surgical cleaning procedure was performed under sedation (Figure 1A,B).

Case 2: Feline, female, domestic short hair (DSH), 6 months, 1.2 kg, belongs to the same group of outdoor cats from Río Verde. She was taken on August 18 to the same veterinary center mentioned before.

The clinical examination showed subcutaneous, sero-bloody erosions located on the face, also associated with deformation of the nasal septum. She had not received any treatment (Figure 1D).

Case 3: Feline, female, DSH, 6 years old, 3.5 kg, indoor, from the city of Punta Arenas. On 8 September, the cat showed during clinical examination a subcutaneous mass on the right costal flank of approximately 8 cm in diameter with an ulcerated and suppurative center. In addition, a left periocular lesion with a moist and scaly appearance was observed (Figure 1F).

The owner reported that previously in another veterinary clinic, they prescribed amoxicillin; however, with this treatment, no improvement was observed.

Each patient underwent feline immunodeficiency and feline leukemia virus tests (Snap FIV/ FeLV^®^ combo test, IDEXX, Westbrook, ME, USA), which were negative. None of these cats or their close contacts had a history of traveling to areas where the *S. brasiliensis* is endemic.

Due to the clinical history and the characteristics of the lesions, a skin biopsy was obtained in all cases. For the procedure, each cat was pre-medicated with Tiletamin 2.5%-Zolazepam 2.5% (Zoletil, Virbac, Santiago, Chile), anesthesia was induced with isoflurane (Baxter Latin America, Santiago, Chile) at 3%, maintaining a sedation dose of 2.5% isoflurane, with a minimum alveolar oxygen concentration of 1.2%. Two biopsies of approximately 7 mm in diameter each were obtained for histopathological and microbiological study [70]. They were later sent to the Laboratory of Anatomic Pathology and Histopathology (CITOVET) and to the Clinical Microbiology and Microbiome Laboratory, of the School of Veterinary Medicine of the Andrés Bello University, both in Santiago, Chile.

## 3. Results

### 3.1. Histopathological Study

The affected areas show an acanthotic to ulcerated epidermis, with variable degrees of exudation and crusting. Throughout the superficial and deep dermis and subcutaneous tissue, there is intense diffuse pyogranulomatous inflammation. There are multiple foci of necrosis surrounded by neutrophil clusters. Prominent blood vessels have proliferation within the leucocyte infiltrate where numerous macrophages are intermingled with neutrophils. Round to elongated yeasts 2 to 3 μm in diameter are present in profusion, both free and within the cytoplasm of macrophages. They are so numerous that they are not difficult to appreciate even by Hematoxylin–Eosin (H/E) Staining. However, their morphological features, corresponding to *Sporothrix* spp., are more prominent and easier to appreciate by Periodic acid-Shiff (PAS) and Grocott stains, where yeasts with frequent budding figures can be seen (Figure 2).

### 3.2. Mycological Study

The phenotypic study was carried out in a BSL-2 biosafety cabinet and based on characteristics of the isolates. Initially, a direct microscopic examination of the sample with Gram staining was performed, where abundant blastoconidia were observed, some of them with single budding (Figure 3A). Then, the tissue sample was minced and seeded on Sabouraud agar glucose (SGA) containing chloramphenicol (0.05 g/L) and cycloheximide (0.4 g/L) (Merk, Rahway, NJ, USA), and incubated at 25 °C and 37 °C for 10 days (Figure 3B). The macro- and micromorphological characteristics of the colony [71,72,73] allowed the presumptive identification of *Sporothrix* spp. (Figure 3C,D).

The genotypic study of the 3 isolated colony included DNA extraction performed with the Quick-DNA Fungal/Bacterial kit (Zimo Research, Irvine, CA, USA), following the manufacturer’s instructions. PCR amplification was performed using universal primers ITS5 and ITS4 [52,74,75,76,77]. The sequences obtained were edited, assembled, and analyzed by comparing the nucleotide similarity using the BLASTn tool of the National Centre for Biotechnology Information (NCBI). The sequences of isolates were successfully confirmed as *S. brasiliensis* (accession number OX416764, OX416765, and OX416766), presenting 100% genetic identity with this species.

In vitro antifungal susceptibility test: The minimum inhibitory concentration (MIC) was determined by broth microdilution, according to document M38-A2. from the Clinical and Laboratory Standards Institute (CLSI) [78]. Each microplate was loaded with RPMI 1640 medium buffered to pH 7.0 with 0.165 mol/L of morpholinopropanesulfonic acid (Sigma-Aldrich, St. Louis, MO, USA).

From the different antifungals obtained in pure powder, concentrations that fluctuated between 0.125 and 64 mg/L were prepared. The drugs tested were amphotericin B (AMB), fluconazole (FLC), itraconazole (ITR), ketoconazole (KET), posaconazole (POS), voriconazole (VRC), and terbinafine (TRB) (Sigma-Aldrich, St. Louis, MO, USA).

Inoculum of each strain of 1–5 × 10^4^ conidia/mL were prepared in sterile saline solution, including *Candida parapsilosis* ATCC 22019 and *C. krusei* ATCC 6258 as reference strains. MICs were determined by visual inspection after 48-72 h of incubation at 35 °C. For AMB, ITR, VRC, and POS, the MICs were the lowest concentrations that 100% inhibited the growth of the fungus. For FLC, KET, and TRB, the MIC was the lowest concentration that resulted in a 50% and 80% reduction in growth, respectively, relative to that of the growth control [78].

The results obtained demonstrated good activity in the most antifungals used against tested tree strains of *S. brasiliensis.* The azoles showing an MIC range of 0.25–1 μg/mL, with the exception that fluconazole showed MIC > 64 μg/mL in all strains. AMB and TRB showed MIC values between 0.5 and 1 μg/mL. The MICs for the control strains were within the expected range, as described by the CLSI guidelines [78].

### 3.3. Treatment and Clinical Evolution

Initial treatment began with itraconazole 10 mg/kg per day for each patient. Once the causative agent was diagnosed, a dose adjustment of 100 mg/kg/day was made. The owners were educated about the zoonotic potential and prognosis for the disease.

Both feral cats (cases 1 and 2) were adopted and are still being treated with itraconazole. In both cases, it has been observed that the dermatological lesions have slowly improved but not completely yet. A noticeable improvement has also been observed in their general condition and body weight. Case 1 is currently still undergoing treatment, until completing 5 months of treatment.

In clinical case 3, a fluctuating behavior on the clinical evolution has been observed. The patient has presented some systemic episodes with fever and loss of body weight. During the antifungal treatment, small new skin lesions have appeared on the face with a tendency to deform the nasal septum. For this reason, potassium iodide (2.5 mg/kg) was also recently indicated, but their effect has not yet been possible to evaluate.

This corresponds to the first report that identifies *S. brasiliensis* in felines from Chile, in this case, from the Magallanes Region. 

## 4. Discussion

The epidemiological characteristic of the clinical cases considered in this study correspond to the description of an outbreak that, according to histopathology, microbiology, and molecular analysis, identifies as *S. brasiliensis.*

The first description of S. *brasiliensis* was made in 2007 in Brazil, and it is currently responsible for more than 90% of the cases of sporotrichosis in humans and felines in that country, where it is considered an endemic mycosis [1,56,76,79,80,81]. Typically, this fungus has been associated with temperate and humid climates; however, this has changed in recent years, as *S. brasiliensis* has been detected in different climatic regions, such as Argentina, where it has also been identified in humans and felines [82].

Among the risk factors associated with feline sporotrichosis, the previously reported research results include cats that live or roam outdoors or have free access to the street, uncastrated males, and territorial behaviors with fights that lead to scratches or bites [6,57,63,64], information that is consistent with the data observed in these cases.

Nodules and subcutaneous ulcers were the more frequently observed lesions in the cases under study; this agrees with previous reports [37,50,55,56,57,58,83]. 

The presumptive identification of the agent was carried out by means of the phenotypic characteristics and the confirmation of the identity, by means of molecular techniques. Currently, the phenotypic characteristic alone is not sufficient to identify species of the genus *Sporothrix* spp., due to the uncertainty of the tests involved. Thus, the use of molecular methodologies is required [37,46,55,71,72,73,74,75,76,77,83,84]; among them, the sequencing of the ITS region of the rDNA is one of the most used [19,72].

The treatment choice in the three cases was itraconazole indicated for 4–6 months. Itraconazole remains the drug of choice for the treatment of feline sporotrichosis and its efficacy in monotherapy has been previously documented [37,55,66,67,83,85,86]. Its in vitro activity against *S. brasiliensis* strains isolated from cats [65,66,87] has also been reported. However, the correlation between in vitro antifungal susceptibility testing and in vivo therapeutic response is not always positive. In this regard, other authors report unsuccessful therapies with this antifungal [67,88,89], which has been associated with the virulence of the species and the spreading of the lesions [33,90].

Some reports recommend itraconazole associated with potassium iodide, for deeper lesions located in the nasal region, which invade the cartilage beneath [58,90]. The general condition of the cat, the presence or absence of respiratory signs, as well as the number, extension, and location of the lesions are factors that can influence the prognosis [9,54,86].

Based on the description of this outbreak, we generate guidelines that provide practical information that can help veterinarians in diagnosing sporotrichosis opportunely and accurately in our country. In addition, programs with a one health focus that include education, responsible pet ownership, and waste containment are required to avoid dissemination into the environment [6,91], thus promoting awareness of the zoonotic potential of this species and the importance for human, animal, and environmental health. For this reason, as soon as the causal agent was identified, the veterinarians working in the area were contacted and invited to a seminar where the problem was exposed and discussed. In addition, a meeting was held with representatives from the Ministry of Health of Magallanes (MINSAL), the Regional Ministerial Secretariat (SEREMI), the Livestock Agricultural Service (SAG), local veterinary services, and human hospitals. As a result, an alert was generated in the region with a distribution of educational material and recommendations for dissemination control actions.

Further ongoing studies will include the search for *S. brasiliensis* in the claws of diseased and clinically healthy animals and from the soil and plants, to help reveal transmission routes. 

## 5. Recommendations

Considering that the lesions present on the skin of cats contain a considerable fungal burden, it is essential to use personal protection elements, such as disposable gloves and aprons, when attending to or having contact with these cases. In cats with continuous secretions and/or multiple skin lesions, the use of an N95 or PFF2 face mask and safety glasses are also recommended [92,93].

## Figures and Tables

**Figure 1 jof-09-00226-f001:**
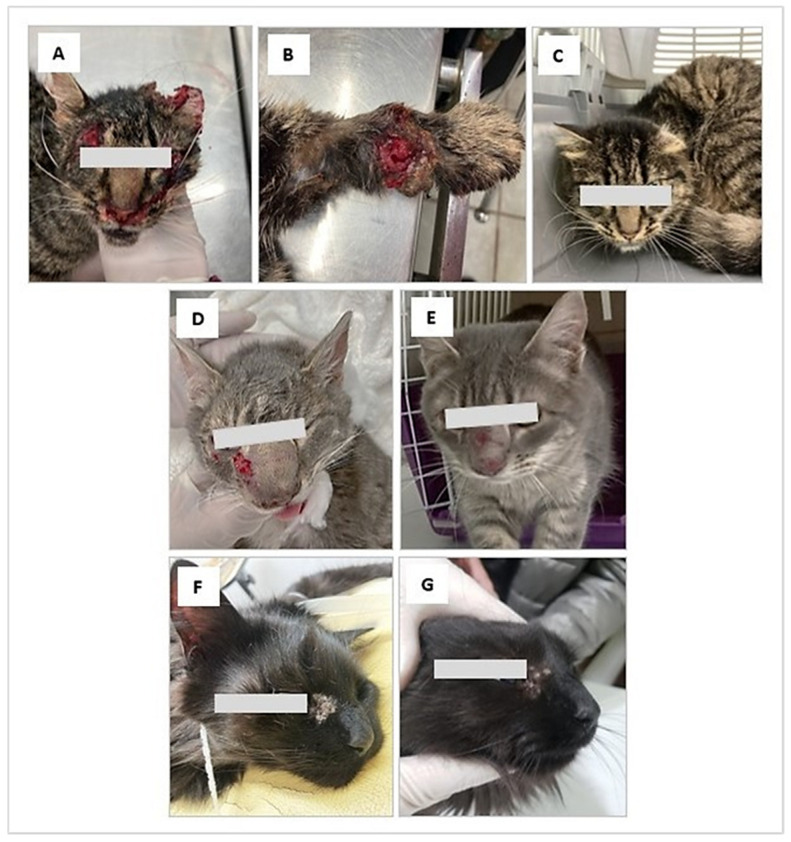
Clinical aspect of feline sporotrichosis. (**A**,**B**) Ulcerated lesions in the cephalic region and forelimb, associated with laceration of the left ear. (**D**) Subcutaneous, sero-bloody erosions located on the face near the eye, with inflammation and deformation of the nasal septum. (**F**): Periocular lesion with a moist and scaly appearance. (**C**,**G**). Clinical appearance of the lesions after treatment (**C**) A remarkable clinical cure is observed after 5 months of treatment with itraconazole (**G**) The evolution of treatment it is favorable after two months with treatment with itraconazole. (**E**) Appearance of lesions after 2 months of treatment with itraconazole and potassium iodide.

**Figure 2 jof-09-00226-f002:**
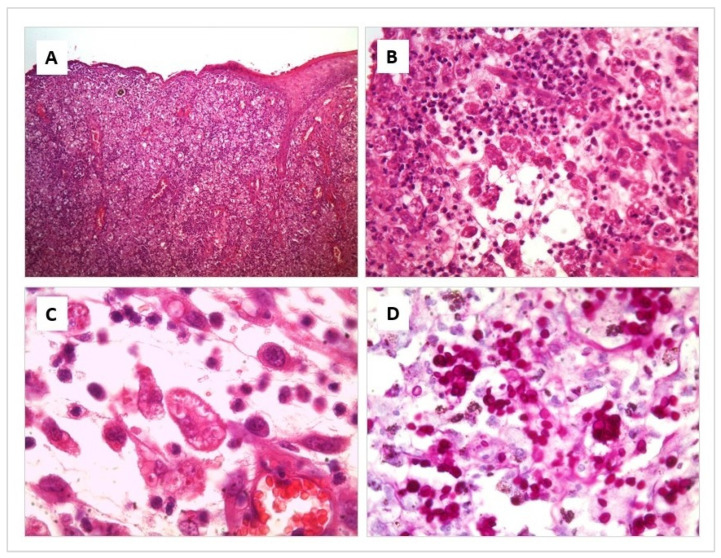
Histological section of a skin lesion from a cat with Sporotrichosis. Hematoxylin-eosin (**A**–**C**) and Periodic acid-Shiff (**D**) stain. (**A**) Diffuse pyogranulomatous inflammation is present throughout the skin layers. (**B**) Multiple foci of necrosis are surrounded by numerous macrophages, intermingled with neutrophils. (**C**) Round to elongated yeasts 2 to 3 μm in diameter are present in profusion, both free and within the cytoplasm of macrophages. (**D**) Numerous yeasts with frequent budding figures can be seen, corresponding to *Sporothrix* spp.

**Figure 3 jof-09-00226-f003:**
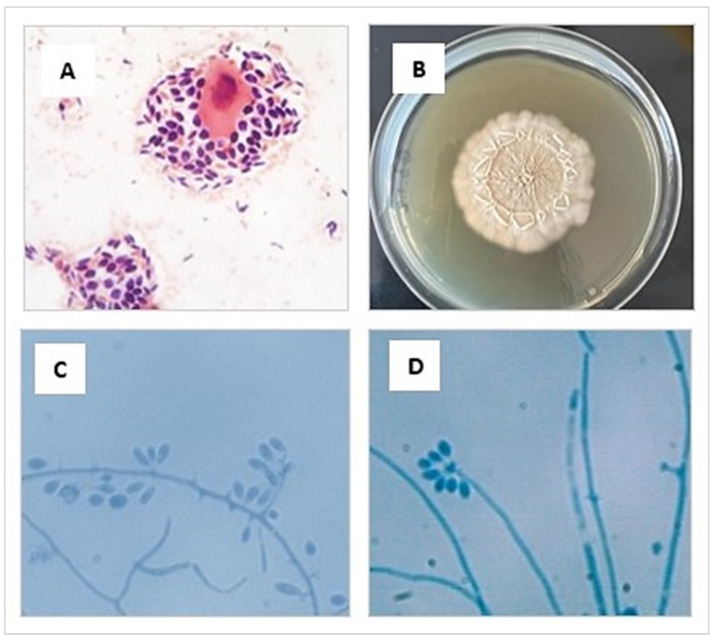
(**A**) Cytologic preparation collected from an ulcerated lesion of the cat’s skin showing an intermediate epithelial cell surrounded by many yeasts, stained using the Gram and observed at 1000× magnification. (**B**) Macromorphology of the colonies of *S. brasiliensis* cultivated in SGA medium for 10 days at 25 °C (**C**,**D**) Preparation made from a microculture of a colony of *S. brasiliensis* grown in SGA. Septate hyaline hyphae are seen, and conidiogenous cells arise from undifferentiated hyphae, forming conidia in groups on small, clustered denticles, stained using the lactophenol cotton blue observed at 1000× magnification.

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
