# Peer review of "Sporotrichosis Outbreak Due to Sporothrix brasiliensis in Domestic Cats in Magallanes, Chile: A One-Health-Approach Study"

_jof, 2023, doi:10.3390/jof9020226_

Round 1

Reviewer 1 Report

As expected, sporotrichosis cases are started to be widespread among South American countries. The article is complete and it can be published in actual form.

Author Response

Dear reviewer, we appreciate all your comments and input. Below we respond to his suggestions:

Reviewer 2 Report

Comments to authors:

Several corrections have to be made in order to achieve a better understanding.

The manuscript would benefit from overall editing for clarity. Some paragraphs are unclear and confusing.

The entire text should be revised by a native English speaker and should be rewritten.

-Write the full name of each strain only the first time, and write the initial of the genus followed by the species afterwards eg. Sporothrix brasiliensis, then S. brasiliensis.

-Pag 1. Title: replace “Sporotrichosis” with “Sporothrix brasiliensis

-Pag. 1-line 17.

Abstract: please review and correct the paragraph “Sporotrichosis is a zoonotic cutaneous…..” Sporothrix luriei and S. globose are not agents of zoonotic sporotrichosis.

-Pag. 1-line 27:  the fungal culture does not confirm a correct and definitive identification.

The definitive identification is confirmed by molecular techniques.

-Pag. 1-line 43 Introduction: delete “considered”

-Pag. 2, line 48

Sporothrix globose is also present in Argentina, Venezuela, (Cordoba et al 2018; Camacho et al, 2018).

-Pag. 2-paragraph lines 62 to 66. Reformulate the paragraph. eg. Line 65: In South America most of the isolates……

Materials and methods

This section is unclear, please add the year when the outbreak occurred, there are some discrepancies with the dates.

Pag. 3. Case description: Reformulate the text, description is unclear and confusing

 DLH, DSH, spell these acronyms.

Pag. 5. Line 185, in “3.1 Histopathological study.” delete the period at the end of the title

Paragraph lines 191, 195. Some sentences are repeated in the legend under Figure 2.

Pag. 6-line 227. Please explain which genotypic characteristic you can observe at the BSL-2.

Line 232: what concentration of chloramphenicol and cycloheximide was used?

Line 234: The macro and micromorphological………allowed the presumptive identification of ….

Pag. 7-paragraph lines 279 to 282,

This section should be exhaustively reviewed and corrected. The scope of the M51-A, CLSI document does not include Sporothrix spp., in addition, no clinical breakpoints have been defined for Sporothrix, thus, categorical interpretation is not possible.

Can you explain and detail the procedure you have employed to determine the in vitro antifungal susceptibility?

What criteria were used to categorize Sporothrix spp., as susceptible/resistant?

Justify the use of caspofungin.

Pag 7. Line 284 in “3.3 Treatment and clinical evolution.” delete the period at the end of the title

 4-Discussion

-The entire section should be reformulated.

-Do not repeat results in this section.

-Do not include the commercial origin of antifungal drugs

Line 344: what is the meaning of the sentence?

 The sentence “we report the first isolation of Sporothrix……” is repeated in too many places.

Line 355: correct “S. brasiliensis

Paragraph line 359 to 377, consider removing this paragraph, maybe you can move it to a “recommendation” section, but it should not be in the “discussion” section.

Author Response

REVISOR 2

Dear reviewer, we appreciate all your comments and input. Below we respond to his suggestions:

The English language of the manuscript was reviewed for its length by a native

Comments to authors:

-Pag 1. Title: replace “Sporotrichosis” with “Sporothrix brasiliensis

R: The correction was made

-Pag. 1-line 17.

Abstract: please review and correct the paragraph “Sporotrichosis is a zoonotic cutaneous…..” Sporothrix luriei and S. globose are not agents of zoonotic sporotrichosis.

R: The correction was made

-Pag. 1-line 27:  the fungal culture does not confirm a correct and definitive identification.

The definitive identification is confirmed by molecular techniques.

R: The sentence was written again

-Pag. 1-line 43

Introduction: delete “considered”

-Pag. 2, line 48

Sporothrix globose is also present in Argentina, Venezuela, (Cordoba et al 2018; Camacho et al, 2018).

R: R: The corrections were made as suggested by the reviewer.

-Pag. 2-paragraph lines 62 to 66. Reformulate the paragraph. eg. Line 65: In South America most of the isolates……

R: The paragraph was reformulated again

Materials and methods

This section is unclear, please add the year when the outbreak occurred, there are some discrepancies with the dates.

R: The correction was made

Pag. 3. Case description: Reformulate the text, description is unclear and confusing

 DLH, DSH, spell these acronyms.

R: The correction was made

Pag. 5. Line 185, in “3.1 Histopathological study.” delete the period at the end of the title

R: The correction was made

Paragraph lines 191, 195. Some sentences are repeated in the legend under Figure 2.

R: The correction was made

Pag. 6-line 227. Please explain which genotypic characteristic you can observe at the BSL-2.

R: The correction was made

Line 232: what concentration of chloramphenicol and cycloheximide was used?

R: The concentration of the products was noted

Line 234: The macro and micromorphological………allowed the presumptive identification of ….

R: The corrections were made as suggested by the reviewer.

Pag. 7-paragraph lines 279 to 282,

This section should be exhaustively reviewed and corrected. The scope of the M51-A, CLSI document does not include Sporothrix spp., in addition, no clinical breakpoints have been defined for Sporothrix, thus, categorical interpretation is not possible.

Can you explain and detail the procedure you have employed to determine the in vitro antifungal susceptibility?

What criteria were used to categorize Sporothrix spp., as susceptible/resistant?

Justify the use of caspofungin.

R: This part of the analysis was again performed by broth microdilution and the details of the methodology and results are expressed in the manuscript.

Pag 7. Line 284 in “3.3 Treatment and clinical evolution.” delete the period at the end of the title

R: The corrections were made as suggested by the reviewer.

4-Discussion

The entire section should be reformulated.

Do no repeat results in this section.

Do not include the commercial origin of antifungal drugs

Line 344: what is the meaning of the sentence?

R: The corrections were made as suggested by the reviewer.

The sentence “we report the first isolation of Sporothrix……” is repeated in too many places.

R: R: The corrections were made as suggested by the reviewer.

Line 355: correct “S. brasiliensis

R: The corrections were made as suggested by the reviewer.

Paragraph line 359 to 377, consider removing this paragraph, maybe you can move it to a “recommendation” section, but it should not be in the “discussion” section.

R: The corrections were made as suggested by the reviewer.

REVISOR 2

Comments to authors:

-Pag 1. Title: replace “Sporotrichosis” with “Sporothrix brasiliensis

R: The correction was made

-Pag. 1-line 17.

Abstract: please review and correct the paragraph “Sporotrichosis is a zoonotic cutaneous…..” Sporothrix luriei and S. globose are not agents of zoonotic sporotrichosis.

R: The correction was made

-Pag. 1-line 27:  the fungal culture does not confirm a correct and definitive identification.

The definitive identification is confirmed by molecular techniques.

R: The sentence was written again

-Pag. 1-line 43

Introduction: delete “considered”

-Pag. 2, line 48

Sporothrix globose is also present in Argentina, Venezuela, (Cordoba et al 2018; Camacho et al, 2018).

R: R: The corrections were made as suggested by the reviewer.

-Pag. 2-paragraph lines 62 to 66. Reformulate the paragraph. eg. Line 65: In South America most of the isolates……

R: The paragraph was reformulated again

Materials and methods

This section is unclear, please add the year when the outbreak occurred, there are some discrepancies with the dates.

R: The correction was made

Pag. 3. Case description: Reformulate the text, description is unclear and confusing

 DLH, DSH, spell these acronyms.

R: The correction was made

Pag. 5. Line 185, in “3.1 Histopathological study.” delete the period at the end of the title

R: The correction was made

Paragraph lines 191, 195. Some sentences are repeated in the legend under Figure 2.

R: The correction was made

Pag. 6-line 227. Please explain which genotypic characteristic you can observe at the BSL-2.

R: The correction was made

Line 232: what concentration of chloramphenicol and cycloheximide was used?

R: The concentration of the products was noted

Line 234: The macro and micromorphological………allowed the presumptive identification of ….

R: The corrections were made as suggested by the reviewer.

Pag. 7-paragraph lines 279 to 282,

This section should be exhaustively reviewed and corrected. The scope of the M51-A, CLSI document does not include Sporothrix spp., in addition, no clinical breakpoints have been defined for Sporothrix, thus, categorical interpretation is not possible.

Can you explain and detail the procedure you have employed to determine the in vitro antifungal susceptibility?

What criteria were used to categorize Sporothrix spp., as susceptible/resistant?

Justify the use of caspofungin.

R: This part of the analysis was again performed by broth microdilution and the details of the methodology and results are expressed in the manuscript.

Pag 7. Line 284 in “3.3 Treatment and clinical evolution.” delete the period at the end of the title

R: The corrections were made as suggested by the reviewer.

4-Discussion

The entire section should be reformulated.

Do no repeat results in this section.

Do not include the commercial origin of antifungal drugs

Line 344: what is the meaning of the sentence?

R: The corrections were made as suggested by the reviewer.

The sentence “we report the first isolation of Sporothrix……” is repeated in too many places.

R: R: The corrections were made as suggested by the reviewer.

Line 355: correct “S. brasiliensis

R: The corrections were made as suggested by the reviewer.

Paragraph line 359 to 377, consider removing this paragraph, maybe you can move it to a “recommendation” section, but it should not be in the “discussion” section.

R: The corrections were made as suggested by the reviewer.

Reviewer 3 Report

General

In general, the work has a very important scientific relevance, showing the presence of S. brasiliensis in an area where until then it had not been detected.

Some details need to be explored in a better way. Everything in the results needs to have a corresponding methodology, which needs to be very clear. I suggest reviewing the order of the methodology, respecting the same order for the results and consequently for the discussion.

More specific details follow below:

Title

In the title, the name “Sporotrichosis brasiliensis” is not correct. I suggest to change the title to: Sporotrichosis outbreak due to Sporothrix brasiliensis in domestic cats at Magallanes, Chile: A one health approach study or A one health approach study in a sporotrichosis outbreak due to Sporothrix brasiliensis in domestic cats at Magallanes, Chile.

Abstract – The first sentence of the abstract (lines 17 and 18) is not corrected. The correct classification of sporotrichosis is the subcutaneous or implantation mycosis. I suggest to change.

In the line 26, change sp to spp.

In the line 27, change the sentence: The fungal culture and the sequence analysis of the ITS region, β-tubulin and elongation factor genes confirmed the diagnosis of the causative agent Sporothrix brasiliensis to: The fungal culture followed to the partial gene sequence and analysis of the ITS region, β-tubulin and elongation factor confirmed the diagnosis of the Sporothrix brasiliensis as the causative agent.

In the line 30, how many cats were treated with potassium iodide associated to itraconazole?

Introduction

The first sentence needs to be corrected as in the abstract.

Change all sp to spp.

Both phases of Sporothrix spp. are cultivated in laboratory. Insert this information in the lines 44 and 45.

In the line 63, change to: highest animal sporotrichosis number.

Materials and Methods

The figure 1 is in the incorrect place. I suggest to put in the results section, not in the methodology.

The methodology of the molecular identification is missing.

These cats have a history of travel to places where S. brasiliensis are endemic? Please, comment this in the text.

Results

I suggest the insertion of a phylogenetic tree in order to demonstrate the Agrupamento of the sequences obtained with the other deposited in the GenBank.

Discussion

Remove the “microrganisms” in the line 360. Change to “considerable number of fungal yeasts” or “considerable fungal burden”.

Author Response

Dear reviewer, we appreciate all your comments and input. Below we respond to his suggestions:

The English language of the manuscript was reviewed for its length by a native

REVISOR 3

Title

In the title, the name “Sporotrichosis brasiliensis” is not correct. I suggest to change the title to: Sporotrichosis outbreak due to Sporothrix brasiliensis in domestic cats at Magallanes, Chile: A one health approach study or A one health approach study in a sporotrichosis outbreak due to Sporothrix brasiliensis in domestic cats at Magallanes, Chile.

R: The tittle was change

Abstract – The first sentence of the abstract (lines 17 and 18) is not corrected. The correct classification of sporotrichosis is the subcutaneous or implantation mycosis. I suggest to change.

R: The correction suggested by the reviewer was made

In the line 26, change sp to spp.

R: The correction suggested by the reviewer was made

In the line 27, change the sentence: The fungal culture and the sequence analysis of the ITS region, β-tubulin and elongation factor genes confirmed the diagnosis of the causative agent Sporothrix brasiliensis to: The fungal culture followed to the partial gene sequence and analysis of the ITS region, β-tubulin and elongation factor confirmed the diagnosis of the Sporothrix brasiliensis as the causative agent.

R: The correction suggested by the reviewer was made

In the line 30, how many cats were treated with potassium iodide associated to itraconazole?

R: Explanation is included

Introduction

The first sentence needs to be corrected as in the abstract.

R: The correction suggested by the reviewer was made

Change all sp to spp.

R: The correction suggested by the reviewer was made

Both phases of Sporothrix spp. are cultivated in laboratory. Insert this information in the lines 44 and 45.

R: The correction suggested by the reviewer was made

In the line 63, change to: highest animal sporotrichosis number.

R: The correction suggested by the reviewer was made

Materials and Methods

The figure 1 is in the incorrect place. I suggest to put in the results section, not in the methodology.

R: The correction suggested by the reviewer was made

The methodology of the molecular identification is missing.

R: The access codes provided by ENA have been added.

We only included sequences from the ITS regions, since it was enough to make an ID.

The idea of doing phylogeny seems excellent to us, however we have reserved it for a manuscript that we are writing, where we will include a greater number of isolates.

This journal suggested us to publish antecedent remains as a clinical case and based on that we deliver the most pertinent information.

These cats have a history of travel to places where S. brasiliensis are endemic? Please, comment this in the text.

R: The correction suggested by the reviewer was made

Results

I suggest the insertion of a phylogenetic tree in order to demonstrate the Agrupamento of the sequences obtained with the other deposited in the GenBank.

R: The idea of doing phylogeny seems excellent to us, however we have reserved it for a manuscript that we are writing, where we will include a greater number of isolates.

This journal suggested us to publish antecedent remains as a clinical case and based on that we deliver the most pertinent information.

Discussion

Remove the “microrganisms” in the line 360. Change to “considerable number of fungal yeasts” or “considerable fungal burden”.

R: The correction suggested by the reviewer was made

Reviewer 4 Report

Particularly interesting research on a fungus of public health importance. I want to highlight some points in the paper.

1) I'd like to ask you why you chose the Giemsa stain.

2) Figure 2 should be marked with the histological finding should be marked with the histological findings where they are observed.

3) I think you have not adequately described all the methods you used. Please describe in detail, particularly from line 222 to 

Author Response

Dear reviewer, we appreciate all your comments and input. Below we respond to his suggestions:

The English language of the manuscript was reviewed for its length by a native

REVISOR 4

Particularly interesting research on a fungus of public health importance. I want to highlight some points in the paper.

  • I'd like to ask you why you chose the Giemsa stain.

R: We do not use Giemsa

  • Figure 2 should be marked with the histological finding should be marked with the histological findings where they are observed.

R: The histological findings were marqued

3) I think you have not adequately described all the methods you used. Please describe in detail, particularly from line 222 to………no more comments are displayed

Round 2

Reviewer 2 Report

Comments to author,

Please review and correct the reference list according to the instructions for authors. In particular, correct reference 78.